# Is there 1.5 million-year old ice near Dome C, Antarctica?

Frédéric Parrenin[1], Marie G. P. Cavitte[2], Donald D. Blankenship[2], Jérôme Chappellaz[1], Hubertus Fischer[3], Olivier Gagliardini[1], Valérie Masson-Delmotte[4], Olivier Passalacqua[1], Catherine Ritz[1], Jason Roberts[5,6], Martin J. Siegert[7], Duncan A. Young[2]

[1]Univ. Grenoble Alpes, CNRS, IRD, IGE, F-38000 Grenoble, France

[2]University of Texas John A. and Katherine G. Jackson School of Geosciences, Institute for Geophysics (UTIG), Austin, USA

[3]Climate and Environmental Physics, Physics Institute, University of Bern, Bern

[4]Laboratoire des Sciences du Climat et de l'Environnement, UMR8212 (CEA-CNRS-UVSQ/IPSL), Gif-Sur-Yvette, France

[5]Australian Antarctic Division, Kingston, Tasmania 7050, Australia

[6]Antarctic Climate & Ecosystems Cooperative Research Centre, University of Tasmania, Hobart, Tasmania 7001, Australia

[7]Grantham Institute, and Department of Earth Science and Engineering, Imperial College, London, UK

*Correspondence to*: F. Parrenin (frederic.parrenin@univ-grenoble-alpes.fr)

**Abstract.** Ice sheets provide exceptional archives of past changes in polar climate, regional environment and global atmospheric composition. The oldest dated deep ice core drilled in Antarctica has been retrieved at EPICA Dome C (EDC), reaching ~800,000 years. Obtaining an older paleoclimatic record from Antarctica is one of the greatest challenges of the ice core community. Here, we use internal isochrones, identified from airborne radar coupled to ice-flow modelling to estimate the age of basal ice along transects in the Dome C area. Three glaciological properties are inferred from isochrones: surface accumulation rate; geothermal flux; and the exponent of the Lliboutry velocity profile. We find that old ice (>1.5 Myr, 1.5 million years) likely exists in two regions: one ~40km south-west of Dome C along the ice divide to Vostok, close to a secondary dome that we name "Little Dome C" (LDC); and a second region named "North Patch" (NP) located 10-30 km north-east of Dome C, in a region where the geothermal flux is apparently relatively low. Our work demonstrates the value of combining radar observations with ice flow modelling to accurately represent the true nature of ice flow, and understand the formation of ice-sheet architecture, in the centre of large ice sheets.

## 1 Introduction

Since around 800,000 years ago, glacial periods have been dominated by a ~100,000 years cyclicity, as documented in multiple proxies from marine, terrestrial and ice core records

(Elderfield et al., 2012; Jouzel et al., 2007; Lisiecki and Raymo, 2005; Loulergue et al., 2008; Lüthi et al., 2008; Wang et al., 2008; Wolff et al., 2006). These data have provided evidence of consistent changes in polar and tropical temperatures, continental aridity, aerosol deposition, atmospheric greenhouse gas concentrations and global mean sea level over numerous glacial cycles. Conceptual models (Imbrie et al., 2011) have been proposed to explain these asymmetric 100,000 yr cycles in response to changes in the configuration of the Earth's orbit and obliquity (Laskar et al., 2004), and involve threshold behavior between different climate states within the

Earth system (Parrenin and Paillard, 2012). The asymmetry between glacial inceptions and terminations may, for example, be due to the slow build-up of ice sheets and their rapid collapse once fully developed due to glacial isostasy (Abe-Ouchi et al., 2013). Observed sequences of events and Earth system modeling studies (Fischer et al., 2010; Lüthi et al., 2008; Parrenin et al., 2013; Shakun et al., 2012) have shown that climate-carbon feedbacks also play a major role in the magnitude of glacial-interglacial transitions.

Critical to our understanding of these 100,000 yr glacial cycles is the study of their onset, during the Mid Pleistocene Transition (MPT, Jouzel and Masson-Delmotte, 2010), which occurred between 1250 and 700 kyr b1950 (thousands of years before 1950 A.D.) (Clark et al., 2006), and most likely during Marine Isotope Stages (MIS) 22-24, around 900 kyr b1950 (Elderfield et al., 2012). Prior to the MPT, marine sediments (Lisiecki and Raymo, 2005) show glacial-interglacial cycles occurring at obliquity periodicities (40 kyr) and with a smaller amplitude. The exact cause for this MPT remains controversial and several mechanisms have been proposed, including: the transition of the Antarctic ice sheet from a wholly terrestrial to a part-marine configuration (Raymo et al., 2006), a hypothesis which is, however, unsupported by long-term simulations (Pollard and DeConto, 2009); a non-linear response to weak eccentricity changes (Imbrie et al., 2011); merging of North American ice sheets (Bintanja and Van de Wal, 2008); changes in sea ice extent (Tziperman and Gildor, 2003); a time varying insolation energy threshold (Tzedakis et al., 2017); a threshold effect related to the atmospheric dust load over the Southern Ocean (Martínez-Garcia et al., 2011); and a long term decrease in atmospheric $CO_2$ concentrations (Berger et al., 1999), the latter hypothesis being challenged by indirect estimates of atmospheric $CO_2$ from marine sediments (Hönisch et al., 2009).

A continuous Antarctic ice core record extending back at least to 1.5 Myr b1950 (million years before 1950 A.D.) would shed new light on the MPT reorganization (Jouzel and Masson-Delmotte, 2010), by providing records of Antarctic temperature, atmospheric greenhouse gas concentrations and aerosol fluxes prior and after the MPT. The opportunity to measure cosmogenic isotopes ($^{10}$Be) would also provide information on changes in the intensity of the Earth's magnetic field, especially during the Jaramillo transition (Singer and Brown, 2002). Retrieving Antarctica's "Oldest Ice" is therefore a major challenge of the ice core science community (Brook et al., 2006). A necessary first step towards this goal is to identify potential drilling sites based on available information on ice-sheet structure and accompanying age modeling (Fischer et al., 2013; Van Liefferinge and Pattyn, 2013).

The maximum age of a continuous ice core depends on several parameters (Fischer et al., 2013). Mathematically, the age χ of the ice at a level $z$ above bedrock can be written as:

$$\chi(z) = \int_z^H \frac{D(z')}{a(z')\tau(z')} dz', \tag{1}$$

where $D(z)$ is the relative density of the material (<1 for the firn and =1 for the ice), $a(z)$ is the accumulation rate (initial vertical thickness of a layer, in m-of-ice year$^{-1}$), $\tau(z)$ is the vertical thinning function, i.e. the ratio of the vertical thickness of a layer in the ice core to its initial vertical thickness at the surface and $H$ is the total ice thickness. Increasing the maximum age $\chi_{max}$ can be obtained by increasing $H$ or by decreasing $a$ or $\tau$. At first glance, one might select a site where $H$ is maximum and $a$ is minimum, but this neglects the importance of $\tau$, notably through basal melting. In general, $\tau$ decreases toward the bed and, in steady-state, reaches the value

$\mu = m/a$ where $m$ is the basal melting. $m$ is therefore a crucial parameter of the problem, as it destroys the bottom of the ice record. As ice is a good insulator, $H$ either increases the ice temperature towards melting for frozen basal ice conditions, or, when melting is present, $m$ increases with $H$ and with the geothermal flux underneath the ice sheet (Fischer et al., 2013). Consequently, "Oldest Ice" sites have better chance to exist where ice is not overly thick as to lead to basal melting (Seddik et al., 2011), yet thick enough to contain a continuous ancient accumulation. The distance of a site to the ice divide is also an important parameter. This distance influences the profile of $\tau$, which is increasingly non-linear right at a dome. Therefore, $\chi_{max}$ can be up to 10 times larger at a dome than a few kilometers downstream (Martín and Gudmundsson, 2012). Moreover, assuming a largely constant ice sheet configuration across glacial cycles, an ice record close to the divide has traveled a shorter horizontal distance and therefore has a better chance of being stratigraphically undisturbed (Fischer et al., 2013).

The depth-age profile in an ice sheet can be obtained using radar observations at VHF frequencies to identify englacial reflections (e.g., Fujita et al., 1999) and trace them as isochrones across the ice sheet (Cavitte et al., 2016; Siegert et al., 1998a). Until now, such analysis has been restricted to the top ~¾ of the ice thickness in East Antarctica. However, depth-age information from internal layers can be used in conjunction with ice flow models and age information from ice cores to extrapolate down to the bed. Radar observations allow estimates of poorly known ice-sheet parameters, such as the geothermal flux (Shapiro and Ritzwoller, 2004) and past changes in the position of ice domes and divides.

The Dome C sector is one of the target areas for the "Oldest Ice" challenge and has a number of distinct benefits over other regions: it has already been heavily surveyed by geophysical techniques (Cavitte et al., 2016; Siegert et al., 1998b; Tabacco et al., 1998), a reference age scale has been developed through the existing ice core work (Bazin et al., 2013; Veres et al., 2013) and it is logistically accessible from the nearby Concordia Station. In this study, we concentrate on airborne radar transects (Fig. 1), which are all related to the EDC ice core. These data resolve the bed (Young et al., 2016) and internal isochrones (Cavitte et al., 2017) and are suitable for Oldest Ice search (Winter et al., 2017). The isochrones are dated up to about 366 kyr b1950 using the most recent AICC2012 chronology established for the EDC ice core (Bazin et al., 2013; Veres et al., 2013). We extrapolate the age of the isochrones toward the bed using an ice flow model in order to identify potential Oldest Ice sites along these transects. We also build maps of surface accumulation rate, geothermal flux and of a linearity parameter of the vertical velocity profile. The spatial and temporal variations of surface accumulation rates are discussed in details in a companion paper (Cavitte et al., 2017).

## 2 Method

We use a 1D pseudo-steady (Parrenin et al., 2006) ice flow model, which assumes that the geometry, the shape of the vertical velocity profile, the ratio $\mu = m/a$ and the relative density profile are constant in time. Only a temporal factor $R(t)$ is applied to both the accumulation rate $a$ and basal melting $m$:

$$a(x,t) = \bar{a}(x)R(t),$$

$$m(x,t) = \bar{m}(x)R(t),$$

(2)

where $\bar{a}(x)$ and $\bar{m}(x)$ are the temporally averaged accumulation and melting rates at a certain point $x$. Under the pseudo-steady assumption, the vertical thinning function is given by:

$$\tau = (1-\mu)\omega + \mu \, , \tag{3}$$

where $\omega$ is the horizontal flux shape function (Parrenin et al., 2006). While there is no physical reason to assume co-variance of basal melting and surface accumulation, comparison with a transient dating model (Parrenin et al., 2007) shows errors of only 6% maximum in the evaluation of the thinning function. Moreover, the fact that there is an analytical expression for the thinning function allows to drastically reduce the computation time, an important factor since the 1D model needs to be applied on many locations and with many different sets of parameters. A steady age $\chi_{\text{steady}}$ is first calculated assuming a steady accumulation $\bar{a}$ and a steady melting $\bar{m}$. Then the real age $\chi$ is calculated using (Parrenin et al., 2006):

$$d\chi_{\text{steady}} = R(t)\,d\chi \, . \tag{4}$$

$R$(t) (Fig. 2) is directly inferred from the accumulation record of the EDC ice core (Bazin et al., 2013; Veres et al., 2013). Beyond 800 kyr, it is assumed to be equal to 1, that is to say that the accumulation before 800 kyr is assumed equal to the average accumulation over the last 800 kyr. The horizontal flux shape function is determined using an analytical expression (Lliboutry, 1979; Parrenin et al., 2007):

$$\omega(\zeta) = 1 - \frac{p+2}{p+1}(1-\zeta) + \frac{1}{p+1}(1-\zeta)^{p+2} \, , \tag{5}$$

where $\zeta = z/H$ is the normalized vertical coordinate (0 at bedrock and 1 at surface) expressed in ice equivalent, and $p$ a parameter modifying the non-linearity of $\omega$ (the smaller $p$, the more non-linear $\omega$). This formulation assumes that there is a negligible basal sliding ratio, as is the case at EDC (Parrenin et al., 2007). This might not be the case elsewhere, but adding a basal sliding term has a similar effect as increasing the $p$ parameter for the top ~¾ of the ice sheet. The age of the ice at any depth is deduced from Eq. (1) using the relative density profile at EDC (Bazin et al., 2013).

To compute the basal melting, we use a simple steady-state 1D thermal model. We solve the heat equation (neglecting the heat production by deformation since there is minimal horizontal shear) as follows:

$$\frac{d}{dz}\left(k_{\text{T}}\frac{dT}{dz}\right) - c\,\rho_i D\,u_z\frac{dT}{dz} = 0 \, , \tag{6}$$

where $T$ is the temperature, $u_z$ is the vertical velocity, $\rho_i$=917 kg m$^{-3}$ is the ice density (Cuffey and Paterson, 2010), $k_{\text{T}}$ (W m$^{-1}$ K$^{-1}$), the thermal conductivity (Cuffey and Paterson, 2010), is given by:

$$k_{\text{T}} = \frac{2\,k_{\text{T}}^i\,D}{3-D} \, , \tag{7}$$

$$k_{\text{T}}^i = 9.828\exp\left(-5.7\times10^{-3}\,T\right) \, , \tag{8}$$

and $c$ (J kg$^{-1}$ K$^{-1}$), the specific heat capacity (Cuffey and Paterson, 2010) is given by:

$$c = 152.5 + 7.122\,T \, . \tag{9}$$

The boundary conditions are:

$$T\big|_{z=H} = T_S \, , \tag{10}$$

$$T\big|_{z=0}=T_f \quad \text{(temperate base),}$$

$$\text{or } -k_\text{T}\frac{dT}{dz}\bigg|_{z=0}=G_0 \quad \text{(cold base),}$$

(11)

where $T_S$=212.74 K is the average temperature at the surface assumed to be equal to the one at Dome C (Parrenin et al., 2013), $G_0$ is the geothermal flux and $T_f$, the fusion temperature is given by Ritz (1992):

$$T_f=273.16-7.4\cdot10^{-8}P-2.4\cdot10^{-8}P' \, ,$$

(12)

where $P'$=$10^6$ Pa is the partial pressure of air and $P$, the pressure, is approximated by the hydrostatic pressure:

$$P=\rho_i g\int_z^H D(z')dz' \, ,$$

(13)

where g=9.81 m s$^{-2}$ is the gravitational acceleration. We used this formula since it gives the best agreement to the measured temperature profile at EDC (Passalacqua et al., 2017). The basal melting is given by:

$$m=\frac{G_0-G}{\rho_i L_f} \quad \text{(temperate base),}$$

$$\text{or } m=0 \quad \text{(cold base),}$$

(14)

where $G$ is the vertical heat flux at the base of the ice sheet and $L_f$=333.5 kJ kg$^{-1}$ is the latent heat of fusion (Cuffey and Paterson, 2010).

To prevent $p$ from being <-1 (Eq. (5) has a singularity for $p$=-1), we write:

$$p=-1+\exp(p') \, .$$

(15)

The values of $a$, $G_0$ and $p'$ are reconstructed using a variational inverse method and using the radar isochrone constraints. The cost function to minimize is formulated using a least-squares expression:

$$S=\sum_{i=1}^N \frac{(\chi_i^{iso}-\chi^{mod}(d_i^{iso}))^2}{(\sigma_i^{iso})^2}+\frac{(p'_{prior}-p')^2}{(\sigma^{p'})^2}+\frac{(G_{0,prior}-G_0)^2}{(\sigma^{G_0})^2} \, ,$$

(16)

where $N$ is the number of isochrones (3≤N≤18, see Table 1 and Fig. 3), $d_i^{iso}$ and $\chi_i^{iso}$ are the depths and ages of the isochrones, respectively, $\sigma_i^{iso}$ is the confidence interval on their age, and $\chi^{mod}$ is the modeled age. $p'_{prior}$=ln(1+1.97) is the *a priori* estimates of $p'$, inferred from the age scale model of the EDC ice core (Parrenin

et al., 2007) and $\sigma^{p'}$=2 is its standard deviation, chosen sufficiently large to allow for a large range of $p$' values. $G_{0,prior}$=51 mW m$^{-2}$ is the a priori estimate of the geothermal flux calculated from satellite magnetic data (Fox Maule et al., 2005; Purucker, 2013), and from analysis of the heat required to maintain melting above subglacial lakes (Siegert and Dowdeswell, 1996). $\sigma^{G_0}$=25 mW m$^{-2}$ is the uncertainty in the geothermal flux (Fox Maule et al., 2005; Purucker, 2013). The total uncertainty of the age of isochrones $\sigma^{iso}$ is composed of: 1) the uncertainty

in the depth of the traced isochrones (Cavitte et al., 2016), transferred in age and 2) the uncertainty of the AICC2012 age of the isochrone at the EDC site.

To solve the least squares problem formulated in Eq. (16), we used a standard Metropolis-Hastings algorithm (Hastings, 1970; Metropolis et al., 1953) with 1000 iterations. This allows not only to obtain a most probable

modelling scenario, but also to quantify the posterior probability distribution, in particular the confidence
intervals or the modeled quantities.

**3 Results and discussions**

In our forward modeling, we used the 1D pseudo-steady assumption. This assumption is very convenient numerically because in this case, there is an analytical expression for the thinning function (Eq. 3). Therefore, there is no need to use a costly Lagrangian scheme, following the trajectories of ice particles. Of course, the
reality is more complex than the pseudo-steady assumption because the temporal variations in melting and accumulation rates are not related and are not the same for each point in space. In Parrenin et al. (2007), we used a more complex age model with a ratio $\mu$ and with an ice thickness allowed to vary in time. The results are very similar with the pseudo-steady model. This is because melting is small compared to the accumulation and the variations in ice thickness are small compared to the total ice thickness. Regarding the spatial pattern of
accumulation, we assumed that it is stable in time, which is roughly confirmed by the inversion of internal layers (Cavitte et al., 2017). Moreover, the 1D assumption dominates the uncertainty since we do not take into account horizontal advection and dome movement. Therefore, we suggest the pseudo-steady assumption is good enough for a 1D model.

An example age profile along the OIA/JKB2n/X45 radar transect (see Fig. 1 for its position) is displayed in Fig.
3. From these profiles, maps of the modeled age at 60 m above the bed, minimum age at 60 m above the bed (at 85% confidence level), the height above the bed of the 1.5 Myr isochrone and temporal resolution at 1.5 Myr are displayed in Fig. 4. We use 60 m above the bed as this is the height at EDC below which the ice becomes disturbed such that it cannot be interpreted stratigraphically (Tison et al., 2015). The modeled basal melting $m$ and inferred steady accumulation rate $a$, geothermal flux $G_0$ and $p$' parameter of the vertical velocity profile are
displayed in Fig. 5.

The bottom age inferred at EDC at 3200 m is 785 kyr, which is remarkably close to the age of ~820 kyr inferred from the analysis of the ice core (Bazin et al., 2013; Veres et al., 2013). These 35 kyr difference represent a depth mismatch of 24 m. This is a confirmation of the method used, despite its assumptions (i.e., 1D, pseudo-steady, Lliboutry velocity profile).

There are two main regions where the basal age is modeled to be older than 1.5 Myr. The first one is situated close to LDC, ~40 km south-west of EDC. In this region that we call LDC Patch (LDCP), the ice thickness is several hundreds of meters lower than at EDC, thus reducing the likelihood of basal melting. The second region is 10-30 km north-east of EDC in the direction of the coast, at a place where the ice thickness is comparable to the one at EDC but with a lower geothermal flux. We call this region "North Patch" (NP). In those two Oldest
Ice spots, the height above the bed of the 1.5 Myr isochrone is modeled to be greater than 150 m. The temporal resolution at 1.5 Myr is ~10 kyr m$^{-1}$, which is sufficient to resolve the main climatic periods (Fischer et al., 2013).

Our LDCP area is generally consistent with Candidate A of Van Liefferinge and Pattyn (Van Liefferinge and Pattyn, 2013) although our area is smaller and constrained to the subglacial highlands under LDC. Van
Liefferinge and Pattyn (2013) did not find a candidate at NP. However, the geothermal heat flux maps they relied on have a lower spatial resolution than the details we examine here. Our model does not predict very old ages for

Candidates B-C-D-E of Van Liefferinge and Pattyn (2013), although the 1D assumption is problematic in those areas since ice particles experienced very different ice thickness conditions along their path.

One possible limitation of our simple ice sheet model is that it does not allow for a layer of accreted ice. We argue that there are no discernable accretion features in the UTIG radargrams, although it is possible that the accretion features do not show up in the basal layer which is difficult to interpret.

We now examine the other variables inferred from the inversion. Basal melting is of course negligible at these two Oldest Ice spots. Melting is, however, significant around EDC (which is consistent with known basal melting at this place), across LDC away and on the bed ridge adjacent to the Concordia Subglacial Trench (called here the Concordia Ridge), consistent with the observation of subglacial lakes (Wright and Siegert, 2012; Young et al., 2016). While it is surprising that basal melting is so large across the ridge of the bed, where the ice thickness is smaller, the 1D assumption is probably invalid in this region, since the ice has been significantly advected horizontally over regions with very different basal conditions (i.e. over the wet-based Concordia Subglacial Trench and then over the adjacent Concordia Ridge which likely has a frozen base). The average surface accumulation rate shows a large-scale north-east to south-west gradient probably linked to a precipitation gradient. It also shows small scale variations linked to surface features and probably due to snow redistribution by wind. The spatial and temporal variations of accumulation are the subject of a companion paper to this study (Cavitte et al., 2017). For the geothermal flux, it should be noted that its reconstruction is only relevant when there is some basal melting (i.e. a temperate base). When the base is cold, its evaluation mainly relies on the prior used for the least squares cost function. Indeed, below the threshold of zero melting, further decreasing the geothermal flux has no effect on the basal melting, and therefore no effect on the modeled age. In the EDC region, the geothermal flux is estimated around 60 mW m$^{-2}$. A high geothermal flux of ~80 mW/m$^2$ is also estimated on the ridge adjacent to the Concordia Subglacial Trench. Again, these results should be taken with caution since they could be an artifact due to the 1D assumption used. The $p$ value inferred at EDC is 2.63, compatible with the value of 1.97±0.93 inferred from the inversion of the EDC age/depth profile (Parrenin et al., 2007). Over the LDC relief, our method infers low $p'$ values, in agreement with the absence of basal melting and therefore basal sliding. This value increases over the Concordia Subglacial Trench and on the south-west side of the LDC bedrock relief, which is probably a sign of increased basal sliding due to the presence of melt water at the ice/bed interface. The very low $p'$ values on the Concordia Ridge adjacent to the Concordia Subglacial Trench are again probably an artifact of the 1D assumption.

**4 Conclusions**

We developed a simple 1D thermo-mechanical model constrained by radar observations to infer the age in an ice sheet. We identified two areas where the age of basal ice should exceed 1.5 Myr. They are located only a few tens of kilometers away from the French-Italian Concordia station, which could provide excellent logistical support for deep drilling. The first area, LDCP, is close to a secondary dome and on a bedrock massif where ice thickness is only ~2700 m. It is located only ~40 km away from the Concordia station in south-westerly direction. The second area, NP, is 10-30 km north-east of Concordia in the direction of the coast. These "oldest ice" candidates will be subject to further field studies to verify their suitability. A 3D model approach would be necessary to study the effect of horizontal advection. Using the shape of the isochrones, which is better

constrained than their absolute age, would bring more light on this problem. The possibility of a layer of stagnant ice should also be investigated. Ultimately, in situ study of the age of the bottom-most ice at these sites will soon be feasible at minimal operational costs using new rapid access drilling technologies (Chappellaz et al., 2012; Schwander et al., 2014), which will provide in-situ measurements to further assess the age of the basal ice and the integrity of the ice core stratigraphy. If successful, this next step will open an exciting opportunity for expanding the EDC records up to a further ~700 kyr back in time, which could help to unveil the mechanisms controlling the last major climate reorganization across the MPT.

## Acknowledgements

We thank the glaciology group at University of Washington and F. Gillet-Chaulet at IGE for helpful discussions. Operational support was provided by the Australian Antarctic Division, the Institut polaire français Paul-Emile Victor (IPEV) and the Italian Antarctic Program (PNRA and ENEA). We thank the staff of Concordia Station and the Kenn Borek Air flight crew. Additional support was provided by the French ANR Dome A project (ANR-07-BLAN-0125). Funding was provided by the French AGIR project "OldestIce", the National Science Foundation grant PLR-0733025 (ICECAP), the Australian Antarctic Project 4346, the G. Unger Vetlesen Foundation and NERC grant NE/D003733/1. This work was supported by the Australian Government's Cooperative Research Centres Programme through the Antarctic Climate & Ecosystems Cooperative Research Centre (ACE CRC). This publication was generated in the frame of Beyond EPICA-Oldest Ice (BE-OI). The project has received funding from the European Union's Horizon 2020 research and innovation programme under grant agreement No. 730258 (BE-OI CSA). It has received funding from the Swiss State Secretariate for Education, Research and Innovation (SERI) under contract number 16.0144. It is furthermore supported by national partners and funding agencies in Belgium, Denmark, France, Germany, Italy, Norway, Sweden, Switzerland, The Netherlands and the United Kingdom. Logistic support is mainly provided by AWI, BAS, ENEA and IPEV. The opinions expressed and arguments employed herein do not necessarily reflect the official views of the European Union funding agency, the Swiss Government or other national funding bodies. This is BE-OI publication number 2 and UTIG contribution ####.

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

| Age (yr) | Uncertainty (yr) |
|---|---|
| 9,989 | 258 |
| 38,106 | 597 |
| 46,410 | 790 |
| 73,367 | 2,071 |
| 82,014 | 1,548 |
| 96,487 | 1,745 |
| 106,247 | 1,822 |
| 121,088 | 1,702 |
| 127,779 | 1,771 |
| 160,372 | 3,581 |
| 166,355 | 3,230 |
| 200,116 | 2,177 |
| 220,062 | 3,019 |
| 254,460 | 4,025 |
| 277,896 | 3,636 |
| 327,339 | 3,053 |
| 341,476 | 4,409 |
| 366,492 | 5,838 |

Table 1: Age and total age uncertainty of the 18 isochrones used in this study.

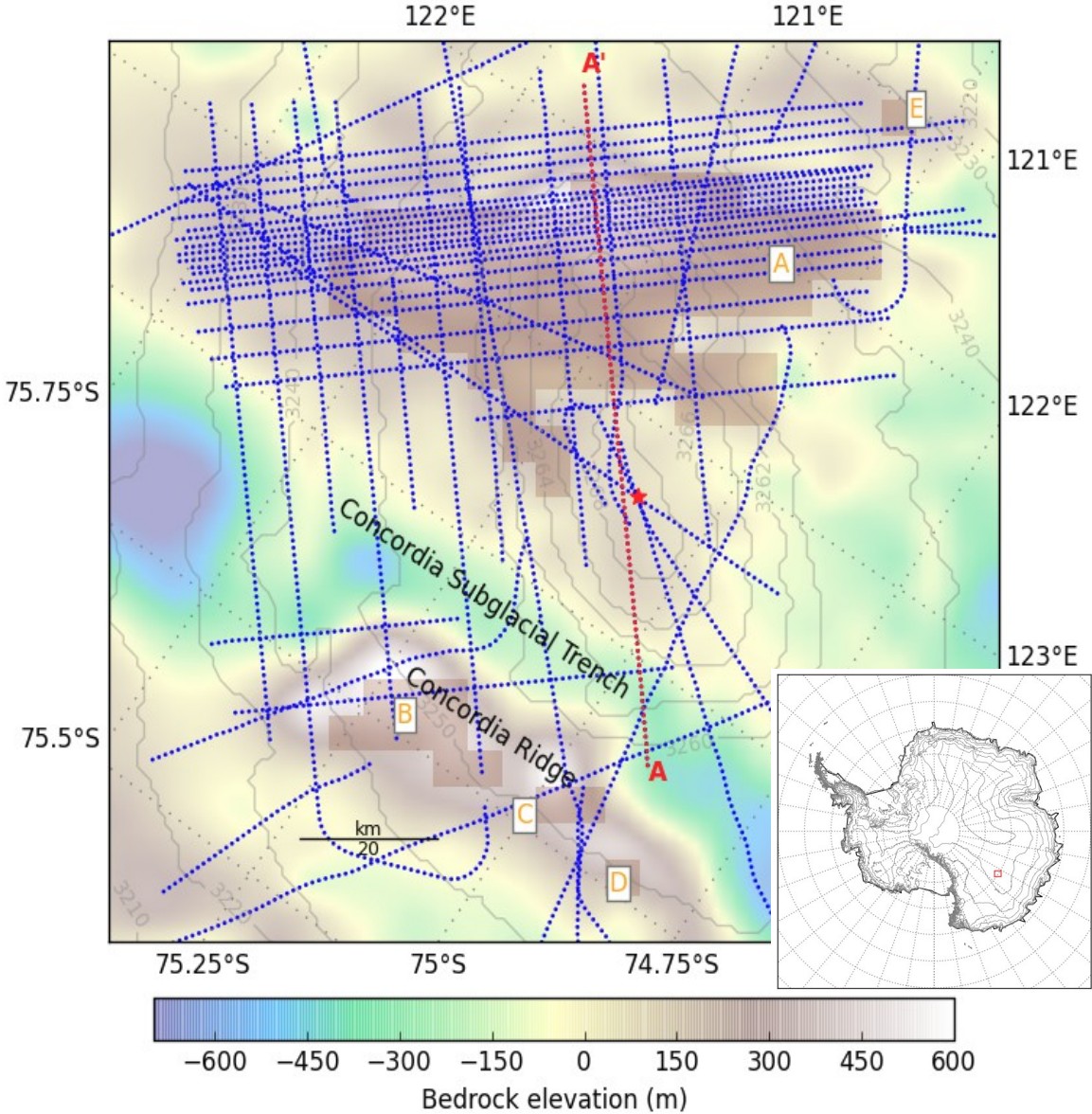

Figure 1: Radar transects used in this study (dotted blue and red lines). The light colour scale represents the bedrock elevation (Fretwell et al., 2013) while the thin grey transparent lines represent the surface elevation (Fretwell et al., 2013). The red square in the inset show the location of the zoomed map around EDC. The red star is the location of the EDC drilling site. The orange squared areas are Oldest Ice candidates from Van Liefferinge and Pattyn (2013). The red dotted line is the OIA/JKB2n/X45 radar line displayed in Figure 3. Note the two candidate sites that we highlight in this study: LDC and NP.

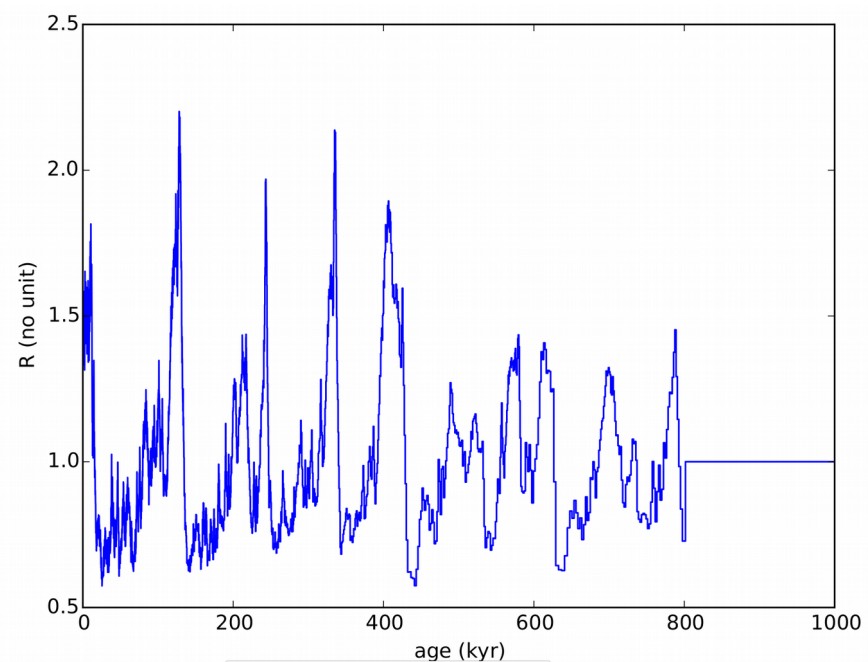

Figure 2: R(t) proportionality factor applied to accumulation and melting rates (see Eq. 2). The plot is cut at 1 Myr for better readability. R(t) is based on the accumulation record at EDC for the last 800 kyr (Bazin et al., 2013; Veres et al., 2013).

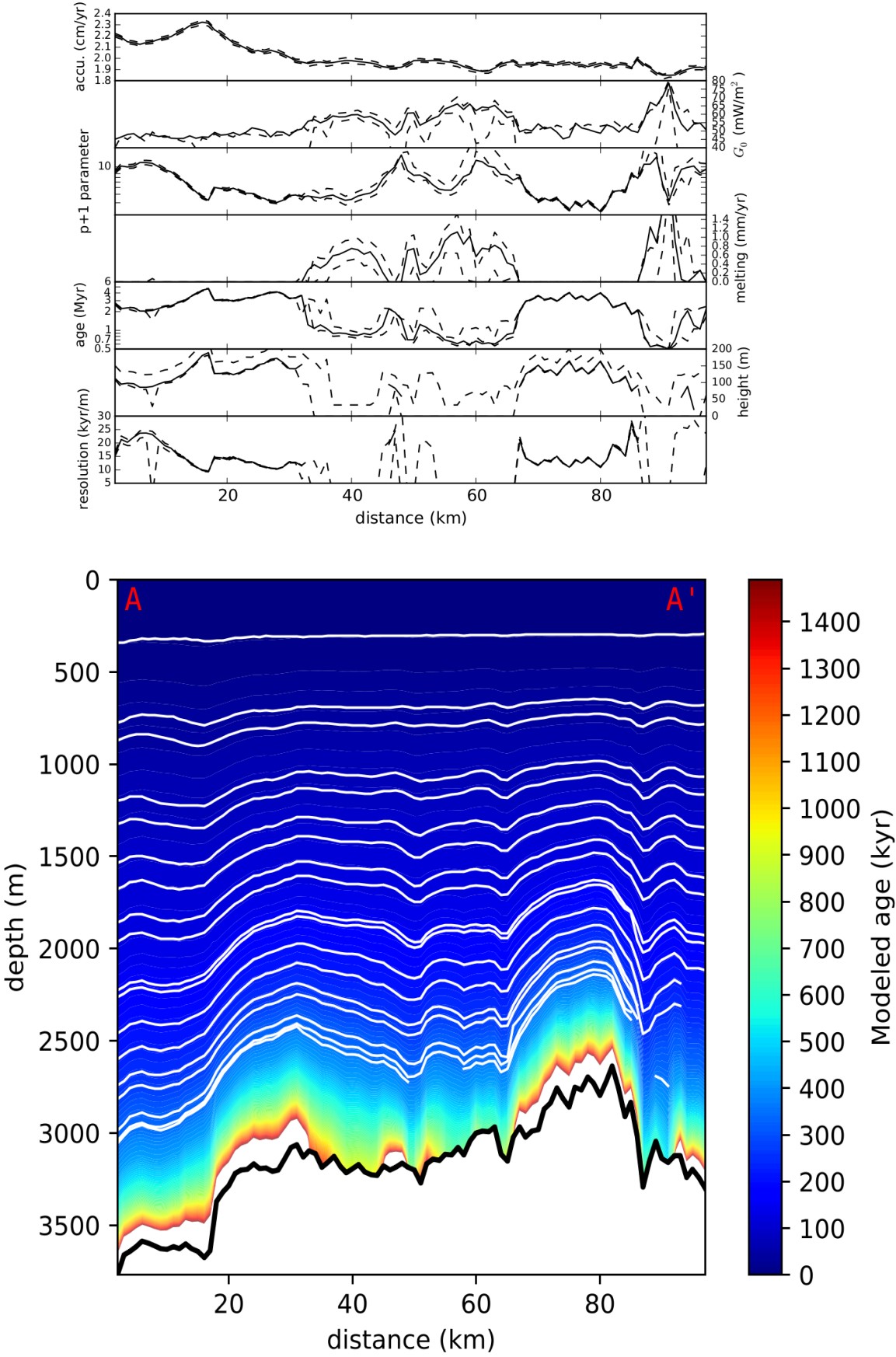

Figure 3: 1D ice flow simulation along the OIA/JKB2n/X45 radar transect (see red dotted line in Fig. 1 for

location) **(TOP)** Various inferred parameters (plain lines) as well as their 15 and 85 percentiles (dashed lines). From top to bottom: average surface accumulation rate, geothermal heat flux, p+1 parameter of the velocity profile, average basal melting, bottom age 60 m above bedrock, height above bed of the 1.5 Myr isochrone and resolution of the 1.5 Myr isochrone. **(BOTTOM)** Modeled age (in colour scale, white is for ages older than 1.5 Myr), together with observed isochrones (in white) and bed (in thick black). Note the two main Oldest Ice

candidates at distance 25 km (NP) and at distance 75 km (LDCP).

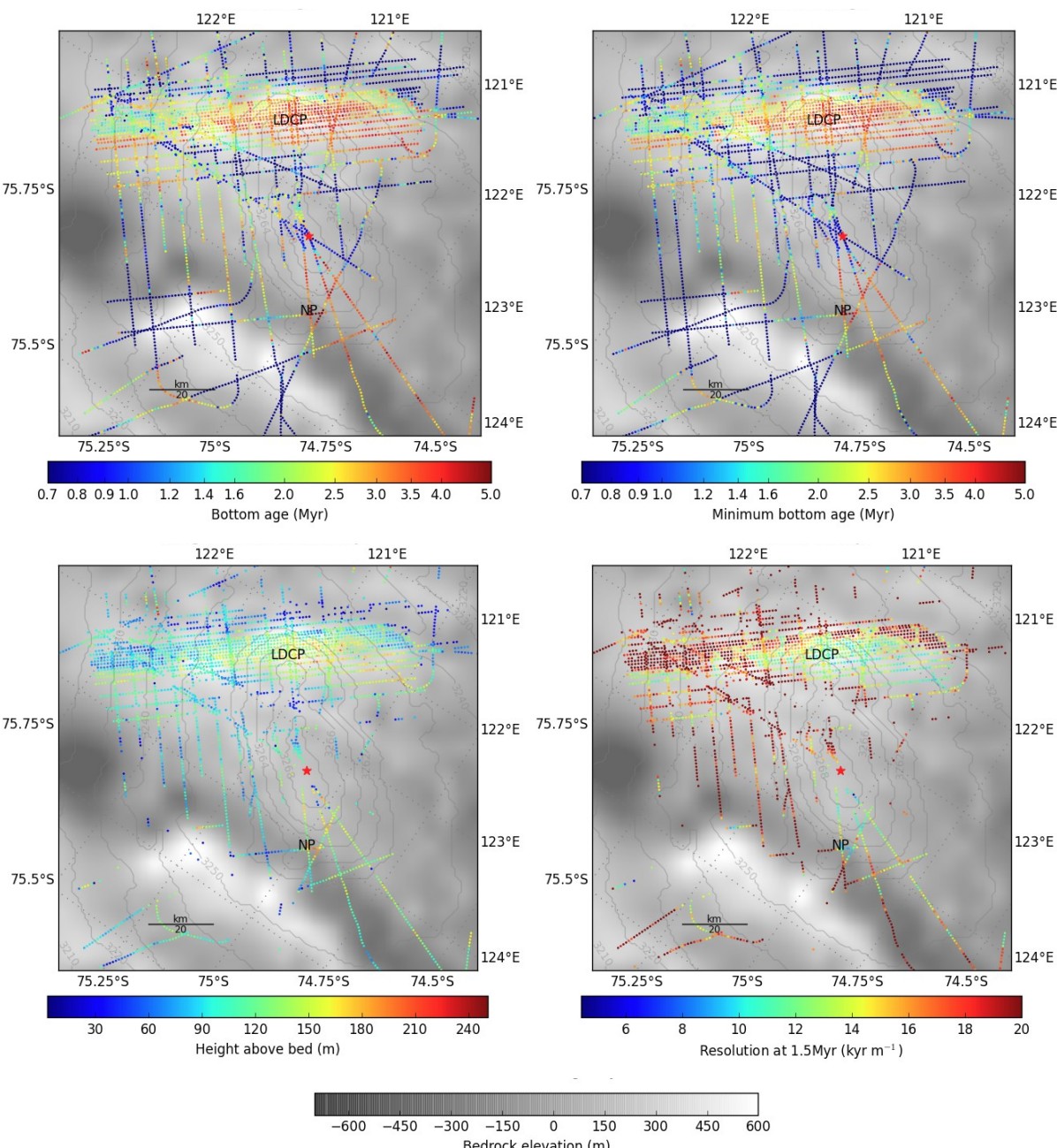

Figure 4: Various bottom age-related variables along the radar transects, in vivid colors. The bedrock and surface elevations (greyscale and isolines, respectively) are shown as in Fig. 1. LDCP and NP are the two old ice patches that we discuss in this study. **(Top-Left)** Modeled bottom age at 60 m above bedrock. **(Top-Right)** Minimum bottom age at 60 m above bedrock with 85% confidence. **(Bottom-Left)** Height above bed of the 1.5 Myr isochrone. **(Bottom-Right)** Temporal resolution for the 1.5 Myr modeled isochrone.

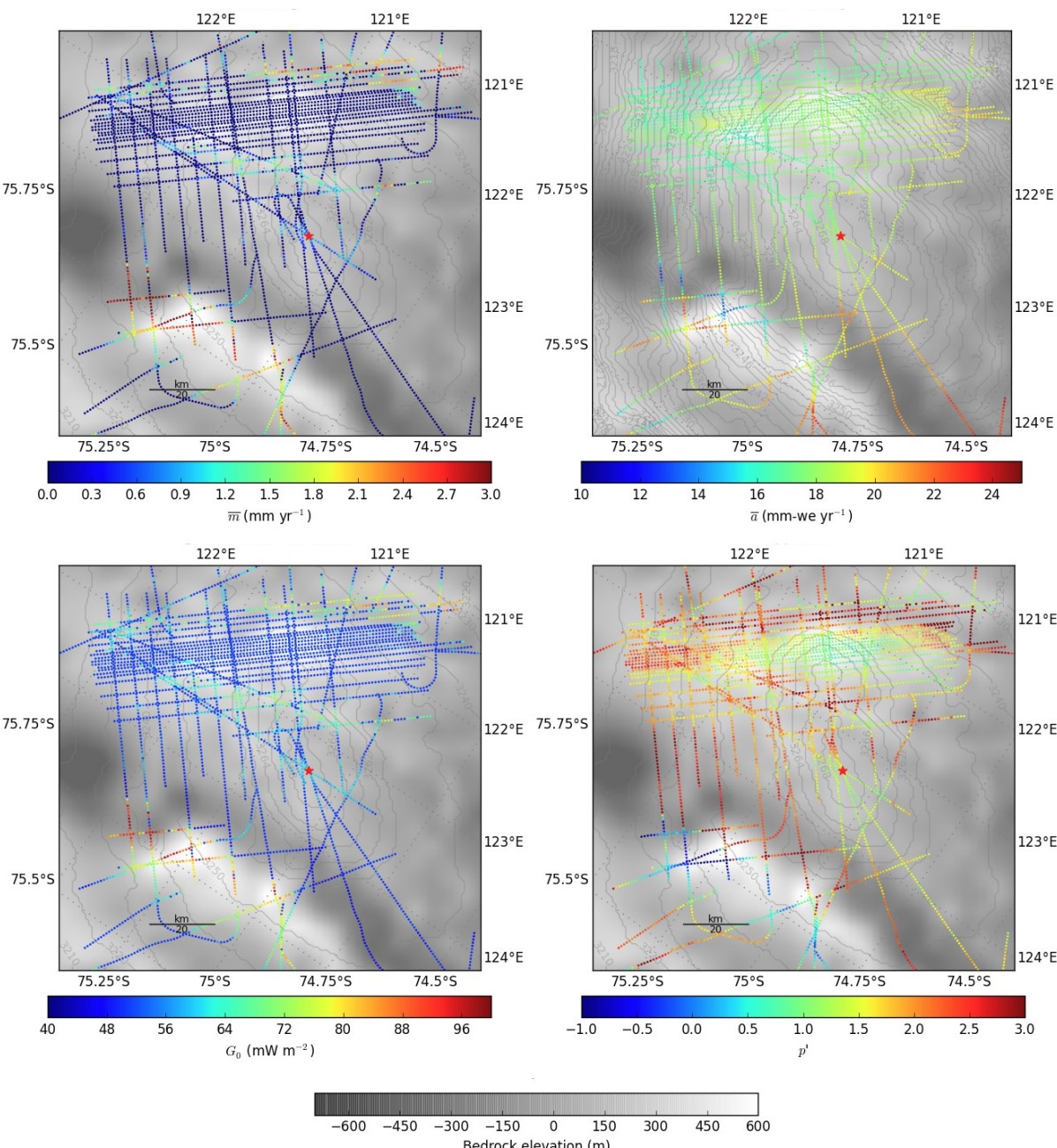

Figure 5: Various variables reconstructed by the inverse method along the radar transects, in vivid colour scale. The bedrock and surface elevations (greyscale and isolines, respectively) are shown as in Fig. 1. **(Top-Left)** Modeled temporally averaged basal melting. **(Top-Right)** Inferred temporally averaged surface accumulation rate. **(Left-Bottom)**. Inferred geothermal flux. **(Left-Right)** Inferred $p'$ vertical velocity parameter.