# Peer review of "Is there 1.5 million-year old ice near Dome C, Antarctica?"

_The Cryosphere, 2017_

## Referee Comment (RC1) · H. Conway (Referee) · 6 Jun 2017

The authors make a compelling case for the existence of >1.5myr-old ice near Dome C; nice work!

They leverage existing ice-core data from nearby EPICA Dome C, airborne radar data, and an established 1-D ice-flow model, [Concerning the radar, you might also cite the recent manuscript: Winter et al. 2017. Comparison of RES measurements and synchronization with the EPICA Dome C ice core. The Cryosphere, 11, 653–668, www.the-cryosphere.net/11/653/2017/, doi:10.5194/tc-11-653-2017].

I have a few questions/comments: 1. Eqn. 2, suggests that $R(t)$ is spatially invariant and that temporal variations in accumulation rate and melt rate are covariant. Is that

the assumption? If so, how does this fit with statements on lines 75-80 that melt rate varies with ice thickness, geothermal flux and accumulation rate? I would have thought that R for melt would vary both spatially and temporally. Perhaps this is related to your model result of high spatial variations of geothermal flux in the region (eg. line 185).

2. Lines 199-201: It is not clear that there are sufficient data constraints (especially histories of accumulation rate and thickness) to construct a realistic 3-D model. However I agree that using new rapid access technologies are the next step.

3. Figures are my main concern: (i) Fig. 1. Text refers to directions south-west of EDC (line 164) and north-east of EDC (line 164). Fig. 1 labels longitudes but not latitudes; at a minimum a north arrow would help orient the reader.

(ii) Fig. 2 shows UTIG profile and gives locations of NP and LDC along this line, and yet Fig 1. shows them offset from the profile by ∼10km and 15km respectively. Please clarify. It would also be useful to show start and end locations of the radar profile in both Figs so the reader does not have to figure it out.

(iii). I struggle with color scales in Figs. 3&4. It is hard t discern gradients in inferred properties that are discussed in the text. Perhaps either changing the range of the color scale, or constructing line plots of a and m would help illustrate the spatial patterns discussed in the text.

(iv) Fig. 3. Line 155 you state why you use 60m above the bed as the transition between disturbed and undisturbed stratigraphy. Hence why do you show results for 150m above the bed (Fig 3, Bottom left)?

(v) Fig. 4: are the accumulation and melt plots the temporally mean values at each point? It would be good to discuss the relevance of the spatial pattern of the p' plot, and I think a plot and discussion of R(t) would also be most informative.

Other editorial comments: Lines 21, 156, 170, 242, 243, and perhaps other places. One of my mentors has pointed out many times that "inverting" or "inverted" means

"to turn upside down"; "to reverse in position, order, direction, or relationship". That is, although you use inverse problem methods, you are not inverting solutions. Rather you are inferring, or deducing solutions.

Line 89: "poorly known" is used twice in the same sentence.

Eqn:13, perhaps better written as m= (Go-G)/(piLf); [when Go>G; 0 when Go<G]

Line 156: "stratigraphically" rather than "stratigraphycally".

---

## Referee Comment (RC2) · R. Greve (Referee) · 19 Jun 2017

This is an interesting study on the hot topic of detecting a suitable site for an "Oldest Ice" core in Antarctica. The 1-D model applied by the authors is rather simple, but it is spiced up significantly by assimilating dated isochrones from airborne radar surveys into the model. The findings are certainly not the final word on the matter (nor do the authors claim that they are), but the paper constitutes a decent step forward along the road.

I only have a few, rather minor comments the authors should consider:

Line 67: "can be written:" -> "can be written as"

Equation (1): This equation is only correct if horizontal flow is neglected. This should

be mentioned already here.

Equation (1): Add a comma after the equation.

Line 74: Add a reference for the equation $\mu = m/a$.

Lines 77/78, "Oldest Ice sites have better chance to exist where ice is not overly thick": I think the work by Seddik et al. (Cryosphere 5, 495-508, 2011, doi: 10.5194/tc-5-495-2011) should be mentioned here. These authors applied a full-Stokes 3-D model (Elmer/Ice) to a $200 \times 200\,\mathrm{km}$ windows around Dome Fuji and found exactly that.

Equation (6): If I am not misled, the symbol $T$ has not been defined.

Line 129: "by (Ritz, 1992)" -> "by Ritz (1992)"

Equation (12): The constant in the equation should be $273.16\,\mathrm{K}$ rather than $273.116\,\mathrm{K}$. The Clausius-Clapeyron constant $7.4 \times 10^{-8}\,\mathrm{K\,Pa^{-1}}$ is the value for pure ice. Using the value $9.8 \times 10^{-8}\,\mathrm{K\,Pa^{-1}}$ for air-saturated glacier ice would have been preferable (Hooke, "Principles of Glacier Mechanics", CUP, 2nd ed., 2005).

Line 130: What exactly is the meaning of $\rho$? The equation for $P$ is only correct if it denotes the depth-averaged density, $\rho = \bar{D}\rho_\mathrm{i}$. Please clarify.

Line 132: "latent heat fusion" -> "latent heat of fusion"

Section 3 (Results and discussions): What I am missing here is a critical discussion of the results against those by Van Liefferinge and Pattyn (2013) [VLP13]. According to Figs. 1 and 3a,b, the authors' LDC area agrees more or less with the candidate "A" by VLP13. However, NP is not close to any of the VLP13 candidates, and the VLP13 candidates "B"-"E" do not look promising in the current study. What are the likely reasons for these discrepancies?

Reference Cavitte et al. (in preparation): I think papers in preparation should not appear in the list of references. They can be mentioned in the main text, though (Cavitte et al., paper in preparation).

References Fischer et al. (2013), Fretwell et al. (2012), Parrenin and Paillard (2012), Young et al. (2016): If available, the respective main papers should be cited rather than the discussion papers.

Reference Lliboutry (1979): "Glacialgeol." -> "Glazialgeol." (German journal title)

References Martin and Gudmundsson (2012), Tison et al. (2015): "The Cryosphere" -> "Cryosphere" (ISO-4 abbreviated title)

Reference Purucker (2013): The URL does not appear properly.

Figure 1, title: "radar-lines" -> "Radar lines"

Figure 2: What is the meaning of the white areas that appear in some places above the bed? Supposedly an age beyond the range of the colour bar? This should either be mentioned in the caption or indicated by a white triangle (or the like) on the top of the colour bar.

Figure 2, caption: "red dotted lien" -> "red dotted line"

Figures 3 and 4: The vivid-coloured dots do not make such a good contrast on the pastel-coloured background. Please consider making the dots bigger for better visibility.

---

## Author Comment (AC1) · 31 Jul 2017

**First of all, we would like to thank you for your constructive and careful review!**

The authors make a compelling case for the existence of >1.5myr-old ice near Dome C; nice work!

**Thank you!**

They leverage existing ice-core data from nearby EPICA Dome C, airborne radar data, and an established 1-D ice-flow model, [Concerning the radar, you might also cite the recent manuscript: Winter et al. 2017. Comparison of RES measurements and synchronization with the EPICA Dome C ice core. The Cryosphere, 11, 653–668, www.the-cryosphere.net/11/653/2017/, doi:10.5194/tc-11-653-2017].

We now cite Winter et al. (2017):

In this study, we concentrate on airborne radar transects (Figure 1), which are all related to the EDC ice core. These data resolve the bed (Young et al., 2016) and internal isochrones (Cavitte et al., 2017) and are suitable for Oldest Ice search (Winter et al., 2017).

**I have a few questions/comments:**

1. Eqn. 2, suggests that R(t) is spatially invariant and that temporal variations in accumulation rate and melt rate are covariant. Is that the assumption? If so, how does this fit with statements on lines 75-80 that melt rate varies with ice thickness, geothermal flux and accumulation rate? I would have thought that R for melt would vary both spatially and temporally. Perhaps this is related to your model result of high spatial variations of geothermal flux in the region (eg. Line 185).

We indeed assume that R(t) is spatially invariant and is the same for both accumulation and melt rates (this is the pseudo-steady assumption). This assumption is very convenient numerically because in this case, there is an analytical expression for the thinning function (Eq. 3). Therefore, there is no need to use a costly Lagrangian scheme. Of course, the reality is more complex than that because the temporal variations in melting and accumulation rates are not related and are not the same for each point in space. In Parrenin et al. (Climate of the Past, 2007, "1D ice flow...", fig. 5), we used a more complex age model with a ratio \mu=m/a and with an ice thickness allowed to vary in time. The results are very similar with the pseudo-steady model (see Figure below), with difference of 6% at maximum mainly due to the ice thickness variations:

This is because melting is small compared to the accumulation and the variations in ice thickness are small compared to the total ice thickness. Moreover, the 1D assumption dominates the uncertainty since we don't take into account horizontal advection and dome movement. Therefore, we reckon the pseudo-steady assumption is good enough for a 1D model. The best would of course be to use a more realistic 3D model, but this is a problem far more complex to tackle... We added the following paragraph at the beginning of the discussion section:

In our forward modeling, we used the 1D pseudo-steady assumption. This assumption is very convenient numerically because in this case, there is an analytical expression for the thinning function (Eq. 3). Therefore, there is no need to use a costly Lagrangian scheme, following the trajectories of ice particles. Of course, the reality is more complex than the pseudo-steady assumption because the temporal variations in melting and accumulation rates are not related and are not the same for each point in space. In Parrenin et al. (2007), we used a more complex age model with a ratio and with an ice thickness allowed to vary in time. The results are very similar with the pseudo-steady model. This is because melting is small compared to the accumulation and the variations in ice thickness are small compared to the total ice thickness. Regarding the spatial pattern of accumulation, we assumed that it is stable in time, which is roughly confirmed by the inversion of internal layers (Cavitte et al., 2017). Moreover, the 1D assumption dominates the uncertainty since we don't take into account horizontal advection and dome movement. Therefore, we suggest the pseudo-steady assumption is good enough for a 1D model.

2. Lines 199-201: It is not clear that there are sufficient data constraints (especially histories of accumulation rate and thickness) to construct a realistic 3-D model. However I agree that using new rapid access technologies are the next step.

We agree that inverting a 3D model is a complex task and it could be an under-constrained problem in general, especially where the radar data are sparse. We believe it is still possible to invert the isochrones with a 3D model if we make relevant assumptions.

3. Figures are my main concern:

(i) Fig. 1. Text refers to directions south-west of EDC (line 164) and north-east of EDC (line 164). Fig. 1 labels longitudes but not latitudes; at a minimum a north arrow would help orient the reader.

We now label more latitudes and longitudes in Figure 1.

(ii) Fig. 2 shows UTIG profile and gives locations of NP and LDC along this line, and yet Fig 1. shows them offset from the profile by  $\sim$ 10km and 15km respectively. Please clarify. It would also be useful to show start and end locations of the radar profile in both Figs so the reader does not have to figure it out.

NP and LDC (now LDCP) refer to broad regions of calculated basal age >1.5 Myr and not to particular points along the radar transect. They are now labeled on Fig. 4 (previously Fig. 3). Start and end of the X45 radar profile are now shown on Fig. 1 and Fig. 3 (previously Fig. 2).

---

## Author Comment (AC2) · 31 Jul 2017

R. Greve (Referee)
greve@lowtem.hokudai.ac.jp

We would like first to thank the reviewer for his careful work and for his constructive comments!

This is an interesting study on the hot topic of detecting a suitable site for an "Oldest Ice" core in Antarctica. The 1-D model applied by the authors is rather simple, but it is spiced up significantly by assimilating dated isochrones from airborne radar surveys into the model. The findings are certainly not the final word on the matter (nor do the authors claim that they are), but the paper constitutes a decent step forward along the road.

Thank you for your generally positive comment.

I only have a few, rather minor comments the authors should consider:
Line 67: "can be written:" -> "can be written as"

Corrected.

Equation (1): This equation is only correct if horizontal flow is neglected. This should be mentioned already here.

With all respect, we disagree here. This equation is also valid with horizontal flow. It just say that the age is the integral from the surface of the number of layers per meter. It has been used many times in previous studies (Ritz, 1992, Parrenin et al., 2001, 2004, just to name a few). In this case, the accumulation is the one at the place and time of snow deposition and the vertical thinning function contains a term related to layer rotation.

Equation (1): Add a comma after the equation.

Corrected.

Line 74: Add a reference for the equation $\mu = m/a$.

This is a new definition and there is no reference for it (to our knowledge).

Lines 77/78, "Oldest Ice sites have better chance to exist where ice is not overly thick": I think the work by Seddik et al. (Cryosphere 5, 495-508, 2011, doi: 10.5194/tc-5-495-2011) should be mentioned here. These authors applied a full-Stokes 3-D model (Elmer/Ice) to a 200 × 200 km windows around Dome Fuji and found exactly that.

Thank you for the relevant reference that we added:

*Consequently, "Oldest Ice" sites have better chance to exist where ice is not overly thick as to lead to basal melting **(Seddik et al., 2011)**, yet thick enough to contain a continuous ancient accumulation.*

Equation (6): If I am not misled, the symbol T has not been defined.

Indeed. This is now corrected:

*We solve the heat equation (neglecting the heat production by deformation since there is minimal horizontal shear):*

$$\frac{d}{dz}\left(k_T \frac{dT}{dz}\right) - c\,\rho_i\,D\,u_z\frac{dT}{dz}=0 \ , \qquad (1)$$

*where **T is the temperature**, $\rho_i=917\ kg\ m^{-3}$ is the ice density (Cuffey and Paterson, 2010), $k_T$ (W m$^{-1}$ K$^{-1}$), the thermal conductivity (Cuffey and Paterson, 2010), is given by:*

Line 129: "by (Ritz, 1992)" -> "by Ritz (1992)"

Corrected.

Equation (12): The constant in the equation should be 273.16 K rather than 273.116 K.

Actually it should be 273.136 (see below).

The Clausius-Clapeyron constant $7.4 \times 10^{-8}$ K Pa$^{-1}$ is the value for pure ice. Using the value $9.8\times10^{-8}$ K Pa$^{-1}$ for air-saturated glacier ice would have been preferable (Hooke, "Principles of Glacier Mechanics", CUP, 2nd ed., 2005).

We use the formula of Ritz et al. (1992) since it gives the best agreement with the EDC temperature profile (see Passalacqua et al., The Cryosphere Discussion, 2017 for more details):

*and $T_f$, the fusion temperature is given by Ritz (1992):*

$$T_f=273.16-7.4\cdot10^{-8}\,P-2.4\cdot10^{-8}\,P' \ , \qquad (2)$$

*where **P'=10$^6$ Pa is the partial pressure of air** and P, the pressure, is approximated by the hydrostatic pressure:*

$$P=\rho_i\,g\int_z^H D(z')\,dz' \ , \qquad (3)$$

*where g=9.81 m s$^{-2}$ is the gravitational acceleration. **We used this formula since it gives the best agreement to the measured temperature profile at EDC (Passalacqua et al., 2017).***

Line 130: What exactly is the meaning of ρ? The equation for P is only correct if it denotes the depth-averaged density, $\rho=\bar{D}\rho_i$. Please clarify.

We made this equation clearer:

*where P, the pressure, is approximated by the hydrostatic pressure:*

$$P=\rho_i\,g\int_z^H D(z')\,dz' \ , \qquad (4)$$

*where g=9.81 m s$^{-2}$ is the gravitational acceleration.*

Line 132: "latent heat fusion" -> "latent heat of fusion"

Corrected.

Section 3 (Results and discussions): What I am missing here is a critical discussion of the results against those by Van Liefferinge and Pattyn (2013) [VLP13]. According to Figs. 1 and 3a,b, the authors' LDC area agrees more or less with the candidate "A" by VLP13. However, NP is not close to any of the VLP13 candidates, and the VLP13 candidates "B"-"E" do not look promising in the current study. What are the likely reasons for these discrepancies?

We added the following paragraph in our discussion:

*Our LDCP area is generally consistent with Candidate A of Van Liefferinge and Pattyn (Van Liefferinge and Pattyn, 2013) although our area is smaller and constrained to the subglacial highlands under LDC. Van Liefferinge and Pattyn (2013) did not find a candidate at NP. However, the geothermal heat flux maps they relied on have a lower spatial resolution than the details we examine here. Our model does not predict very old ages for Candidates B-C-D-E of Van Liefferinge and Pattyn (2013), although the 1D assumption is problematic in those areas since ice particles experienced very different ice thickness conditions along their path.*

Reference Cavitte et al. (in preparation): I think papers in preparation should not appear in the list of references. They can be mentioned in the main text, though (Cavitte et al., paper in preparation).

This reference is now in *The Cryosphere Discussions.*

References Fischer et al. (2013), Fretwell et al. (2012), Parrenin and Paillard (2012), Young et al. (2016): If available, the respective main papers should be cited rather than the discussion papers.

Corrected.

Reference Lliboutry (1979): "Glacialgeol." -> "Glazialgeol." (German journal title)

Corrected.

References Martin and Gudmundsson (2012), Tison et al. (2015): "The Cryosphere" -> "Cryosphere" (ISO-4 abbreviated title)

Corrected.

Reference Purucker (2013): The URL does not appear properly.

Corrected.

Figure 1, title: "radar-lines" -> "Radar lines"

Title has been removed.

Figure 2: What is the meaning of the white areas that appear in some places above the bed? Supposedly an age beyond the range of the colour bar? This should either be mentioned in the caption or indicated by a white triangle (or the like) on the top of the colour bar.

We now mentioned it in the caption:

*Modeled age (in colour scale, **white is for ages older than 1.5 Myr**) along the OIA/JKB2n/X45 radar transect (see red dotted line in Figure 1 for location), together with observed isochrones (in white).*

Figure 2, caption: "red dotted lien" -> "red dotted line"

Corrected.

Figures 3 and 4: The vivid-coloured dots do not make such a good contrast on the pastel-coloured background. Please consider making the dots bigger for better visibility.

For better visibility, we now use a grey scale for the bedrock and we made the dots bigger:

---

## Author Response (AR1)

St Martin d'Hères, July 31th 2017.

Frédéric Parrenin
Institut des Géosciences de l'Environnement
54, rue Molière
38400 St Martin d'Hères
frederic.parrenin@univ-grenoble-alpes.fr
+33.4.76.82.42.65.

Dear Editor,

Please find attached our revised manuscript entitled "Is there 1.5 million-year old ice near Dome C, Antarctica?" for publication in *The Cryosphere*.

The reviews were very helpful and we think we clarified the manuscript thanks to them.

Please feel free to contact me if you need any further information.

Yours sincerely,

Frédéric Parrenin

[revised manuscript text omitted]
 averagedverted steady surface accumulation rate. **(Left-Bottom).** Inferredverted geothermal flux. **(Left-Right)** Inferredverted $p'$ vertical velocity parameter.

---

## Author Response (AR2)

Dear authors

The revised manuscript satisfactory addressed most of reviewer's concerns, but not all (see list below). Therefore, I request a minor revision and will review the next manuscript by myself.

I would like to remind the authors to strictly follow the guidance provided at the journal web site. Authors are supposed to provide point-to-point responses and a marked manuscript.

The author's response (also final author comment in the public discussion) in case of "minor" or "major" revisions must be submitted as one separate *.pdf file (indicating page and line numbers), structured in a clear and easy-to-follow sequence: (1) comments from referees/public, (2) author's response, and (3) author's changes in manuscript. Regarding author's changes, a marked-up manuscript version (track changes in Word, latexdiff in LaTeX) converted into *.pdf including the author's response must be provided.

Dear Editor,

Thank you very much for your careful reading of our revised manuscript as well as of our answers to the reviewers. It helped us make our manuscript even clearer. If there are still unclear points, please don't hesitate to contact us.

Authors' responses to the following points are unsatisfactory.

Reviewer #1's comments (Howard Conway) on (1) R(t),

*1. Eqn. 2, suggests that R(t) is spatially invariant and that temporal variations in accumulation rate and melt rate are covariant. Is that the assumption? If so, how does this fit with statements on lines 75-80 that melt rate varies with ice thickness, geothermal flux and accumulation rate? I would have thought that R for melt would vary both spatially and temporally. Perhaps this is related to your model result of high spatial variations of geothermal flux in the region (eg. Line 185).*

Yes, spatial invariance of R(t) and co-variance of surface accumulations and basal melting is an assumption, as we answered already. This is what we call the pseudo-steady state. There is no physical argument to support the fact that temporal variations in melting and accumulation are the same. However, 1) the difference with a model without this assumption is small, see the graph below comparing the thinning function in Parrenin et al. (CP, 2007) using a transient model, and the thinning function of a pseudo-steady model; 2) this assumption is very convenient since in this case, there is an analytical expression for the thinning function, hence no need to develop a complex model with lagrangian tracers transport; 3) this assumption makes the codes several orders of magnitude faster, thus maintaining a manageable computation time.

We could develop a transient model with a lagrangian scheme, but it would take months of additional work, with a computation time probably difficult to manage and with a result probably very similar to the one with the pseudo-steady assumption. So we decided that our current results, which are already an advance with respect to the 'state-of-the-art' knowledge of OldestIce around Concordia, can already be published.

[Figure]

*Under the pseudo-steady assumption, the vertical thinning function is given by:*

$$\tau = (1-\mu)\omega + \mu \ ,\tag{1}$$

*where $\omega$ is the horizontal flux shape function (Parrenin et al., 2006).* **While there is no physical reason to assume co-variance of basal melting and surface accumulation, comparison with a transient dating model (Parrenin et al., 2007) shows errors of 6% maximum in the evaluation of the thinning function. Moreover, the fact that there is an analytical expression for the thinning function allows to drastically reduce the computation time, an important factor since the 1D model needs to be applied on many locations and with many different sets of parameters.** *A steady age $\chi_{steady}$ is first calculated assuming a steady accumulation $\bar{a}$ and a steady melting $\bar{m}$. Then the real age $\chi$ is calculated using (Parrenin et al., 2006):*

(2) #3-ii (locations of NP and LDCP),

*(ii) Fig. 2 shows UTIG profile and gives locations of NP and LDC along this line, and yet Fig 1. shows them offset from the profile by ~10km and 15km respectively. Please clarify. It would also be useful to show start and end locations of the radar profile in both Figs so the reader does not have to figure it out.*

We now put NP and LDCP right on the X45 profile shown in Fig. 3. Note that, as we explained, NP and LDCP refer to broad regions of calculated basal age >1.5 Myr and not to particular points along the radar transect. They are now labeled on Fig. 4 (previously Fig. 3), since Fig. 4 is a result figure while Fig. 1 is an introductory figure. Start and end of the X45 radar profile are now shown on Fig. 1 and Fig. 3 (previously Fig. 2) using the A and A' labels.

and (3) #3-iii (Figs 3 and 4). For (3), please see my comment below.

*(iii). I struggle with color scales in Figs. 3&4. It is hard t discern gradients in inferred properties that are discussed in the text. Perhaps either changing the range of the color scale, or constructing line plots of a and m would help illustrate the spatial patterns discussed in the text.*

As we explained, we changed the colormap of the bedrock elevation to greyscale and we increased the dot size for a better readability, for example:

[Figure]

Moreover, we now added line plots of various variables in the top panel of Figure 3 (formally Figure 2).

Reviewer #2's comments (Ralf Greve) on (1) Equation 1 (neglecting lateral flow)

*Equation (1): This equation is only correct if horizontal flow is neglected. This should be mentioned already here.*

It is always possible to define a vertical thinning function as the ratio of the vertical thickness of a layer in an ice sheet to its vertical thickness when it was at surface. As we explained, Equation (1) just states that the age is the integral from the surface of the number of layers per unit vertical distance. This equation has been used many times in 2.5D ice flow dating model (Ritz, 1992; Parrenin et al., 2001; Parrenin et al., 2004, just to name a few).

and (2) Clausius-Clapeyron constant.

*The Clausius-Clapeyron constant $7.4 \times 10^{-8}$ K Pa$^{-1}$ is the value for pure ice. Using the value $9.8 \times 10^{-8}$ K Pa$^{-1}$ for air-saturated glacier ice would have been preferable (Hooke, "Principles of Glacier Mechanics", CUP, 2nd ed., 2005).*

As we explained, we use the formula of Ritz (1992), which takes into account the partial pressure of the air in the ice (which is not saturated in ice sheets by the way), and which gives the best agreement with the measured temperature profile (Passalacqua et al., TCD, 2017). This is a text extracted from Passalacqua et al. (TCD, 2017), where this is explained in details:

At the ice/bed interface, the thermal conditions depend on whether the pressure melting point is reached or not. For thawing ice, the temperature simply equals the melting temperature. The melting temperature of pure ice linearly depends on the pressure following a Clapeyron law, for which

180  the corresponding coefficient is $\mathcal{B} = 0.074\,\mathrm{K\,Pa^{-1}}$. For glacier ice, a coefficient of $0.098\,\mathrm{K\,Pa^{-1}}$ was derived from measurements in Blue Glacier, Washington (Cuffey and Paterson, 2010; Harrison, 1972), accounting for the presence of saturated air dissolved in the ice. However, in ice sheets,

the air content is about $0.1\,\mathrm{cm^3/g}$ (Martinerie et al., 1992), whereas the nitrogen saturates in ice at $\sim 2.6\,\mathrm{cm^3\,g^{-1}}$ under a pressure of $27\,\mathrm{MPa}$ (Wiebe et al., 1932). The air is far from saturation for ice

185  sheets. Only the partial pressure of air in the ice $P'$ should be accounted for, so that the dependence of the melting temperature $T_m$ on the pressure $P$ (in $\mathrm{MPa}$) and $P'$ is expressed as (Ritz, 1992):

$$T_m = 273.16 - 0.074 \cdot P - 0.024 \cdot P' \tag{9}$$

In ice sheets, $P'$ is of the order of $1\,\mathrm{MPa}$. This is an unusual choice for such an important parameter, but we argue that Eq.(9) is consistent with the temperature profile of the EPICA Dome C ice

190  core, where the deepest measured temperature was $270.05\,\mathrm{K}$ at $3223\,\mathrm{m}$, $50\,\mathrm{m}$ above the bedrock. The temperature profile can be extrapolated to the bedrock (similar to Dahl-Jensen et al. (2003) at North GRIP) to $271.04\,\mathrm{K}$. The melting temperature computed with $\mathcal{B} = 0.098\,\mathrm{K.Pa^{-1}}$ would be $0.8\,\mathrm{K}$ too low, whereas, it is found to be $270.96\,\mathrm{K}$ with Eq.(9). As the ice moves very slowly in the

Additional comments from the editor

L131: Clarify whether co-variance of surface accumulation and basal melting is an assumption or not. Are there any arguments to support this assumption? This is the point brought by the reviewer #1 and I think that the authors did not respond to this comment.

See our answer above.

L151: Is U_z defined?

Indeed, it was not defined. We now define it:

*To compute the basal melting, we use a simple steady-state 1D thermal model. We solve the heat equation (neglecting the heat production by deformation since there is minimal horizontal shear) as follows:*

$$\frac{d}{dz}\left(k_T \frac{dT}{dz}\right) - c\,\rho_i\,D\,u_z \frac{dT}{dz} = 0 \ , \qquad\qquad (6)$$

*where T is the temperature, $u_z$ is the vertical velocity, $\rho_i=917$ kg m$^{-3}$ is the ice density (Cuffey and Paterson, 2010), $k_T$ (W m$^{-1}$ K$^{-1}$), the thermal conductivity (Cuffey and Paterson, 2010), is given by:*

L196: New Figure 4 shows height of the 1.5Ma isochrone, not the age at 150 m above bed.

Corrected:

*An example age profile along the OIA/JKB2n/X45 radar transect (see Fig. 1 for its position) is displayed in Fig. 3. From these profiles, maps of the modeled age at 60 m above the bed, minimum age at 60 m above the bed (at 85% confidence level), **the height above the bed of the 1.5 Myr isochrone** and temporal resolution at 1.5 Myr are displayed in Fig. 4. We use 60 m above the bed as this is the height at EDC below which the ice becomes disturbed such that it cannot be interpreted stratigraphically (Tison et al., 2015). The modeled basal melting m and inferred steady accumulation rate a, geothermal flux $G_0$ and p' parameter of the vertical velocity profile are displayed in Fig. 5.*

L209-211: The text mentions the age of ice 150 m above the bed, which is no longer presented in Fig. 4.

Corrected:

*There are two main regions where the basal age is modeled to be older than 1.5 Myr. The first one is situated close to LDC, ~40 km south-west of EDC. In this region that we call LDC Patch (LDCP), the ice thickness is several hundreds of meters lower than at EDC, thus reducing the likelihood of basal melting. The second region is 10-30 km north-east of EDC in the direction of the coast, at a place where the ice thickness is comparable to the one at EDC but with a lower geothermal flux. We call this region "North Patch" (NP). In those two Oldest Ice spots, **the height above the bed of the 1.5 Myr isochrone is modeled to be greater than 150 m**. The temporal resolution at 1.5 Myr is ~10 kyr m$^{-1}$, which is sufficient to resolve the main climatic periods (Fischer et al., 2013).*

L229: Does "the ridge" refer the "Concordia Ridge"? Clarify please.

We clarified:

Melting is, however, significant around EDC (which is consistent with known basal melting at this place), across LDC away and on the bed ridge adjacent to the Concordia Subglacial Trench **(called here the Concordia Ridge)**, consistent with the observation of subglacial lakes (Wright and Siegert, 2012; Young et al., 2016). While it is surprising that basal melting is so large across the ridge of the bed, where the ice thickness is smaller, the 1D assumption is probably invalid in this

region, since the ice has been significantly advected horizontally over regions with very different basal conditions (i.e. over the wet-based Concordia Subglacial Trench and then over the adjacent **Concordia Ridge** which likely has a frozen base).

L231-232: Do the radar data show any of accretion features there?

There are no discernible accretion features in the UTIG radargram, but it does not mean that there is no accretion, since the basal layer is difficult to interpret in radargrams. We added the following paragraph:

*Our LDCP area is generally consistent with Candidate A of Van Liefferinge and Pattyn (Van Liefferinge and Pattyn, 2013) although our area is smaller and constrained to the subglacial highlands under LDC. Van Liefferinge and Pattyn (2013) did not find a candidate at NP. However, the geothermal heat flux maps they relied on have a lower spatial resolution than the details we examine here. Our model does not predict very old ages for Candidates B-C-D-E of Van Liefferinge and Pattyn (2013), although the 1D assumption is problematic in those areas since ice particles experienced very different ice thickness conditions along their path.*

***One possible limitation of our simple ice sheet model is that it does not allow for a layer of accreted ice. We argue that there are no discernable accretion features in the UTIG radargrams, although it is possible that the accretion features do not show up in the basal layer which is difficult to interpret.***

*We now examine the other variables inferred from the inversion. Basal melting is of course negligible at these two Oldest Ice spots.*

Also, as we wrote in the conclusion, we will investigate the possibility to add a layer of stagnant ice in the inversion.

L236-239: Authors noted that geothermal flux estimates are less reliable when the bed is not melting. Then, does it make more sense to show the estimated geothermal flux in Fig. 5 only when the bed is melting?

We do not think so, since with don't know for sure where the bed is melting and where it is not. Sometimes, there is 49% chances the bed is melting and 51% chances the bed is not melting. In this case, our model is still able to change the optimal value and the confidence interval of the geothermal flux significantly, which is worth showing (see our new panel in Fig. 3).

L256: Is this LDC or LDCP?

Sorry, this is LDCP:

*The first area, LDC**P**, is close to a secondary dome and on a bedrock massif where ice thickness is only ~2700 m.*

L260: do you mean "Using the shape of the isochorones"?

Yes, we meant that:

*Using the shape of the **isochrones**, which is better constrained than their absolute age, would bring more light on this problem.*

Figure 1: label LDC and NP in the figure. These labels were included in the first manuscript but not in the revised manuscript.

We decided to label LDCP and NP on Figure 4, since Figure 1 is an introductory figure and Figure 4 is a result figure.

Figure 1 caption: "radar transect", not lines.

Corrected:

*Radar **transects** used in this study (dotted blue and red lines).*

Figure 2 caption: Is it better to say "Proportionality factor R(t) applied to accumulation and melting rates (see Eq. 2)." ?

Changed according to your suggestion:

*R(t) proportionality factor applied to accumulation and melting rates **(see Eq. 2)**.*

Figure 3: Mention in the caption or label in the figure that the black curve represents the bed. NP and LDCP are not sites but more like small regions (if I understand correctly). The caption should be updated to clarify this point. The current caption can be read that LDCP and NP are sites exactly on the profile. Also please add a location of the EDC core site (vertical bar in the figure?). I suggest showing the locations of LDCP and NP as well using vertical bars.

Thank you for your suggestion:

*Modeled age (in colour scale, white is for ages older than 1.5 Myr) along the OIA/JKB2n/X45 radar transect (see red dotted line in Fig. 1 for location), together with observed isochrones (in white) **and bed (in thick black)**.*

Figure 3: Add several panels (line plots) above the age diagram to more clearly show some parameters shown in Figs. 4 and 5: (1) bottom age, (2) minimum bottom age, (3) height of 1.5Ma ice above the bed, (4) temporal resolution of 1.5Ma ice, and (5) inferred geothermal flux. I see that

the authors revised Figs. 4 and 5 to respond reviewer's comment. However, it is still hard to read these properties form the maps.

We now have a top panel in Fig. 3 with 7 variables as well as their 15% and 85% percentile: accumulation, geothermal flux, p+1, melting, bottom age, height above bed and resolution of the 1.5 Myr isochrone.

[Figure]

[Figure]

Figure 3: **1D ice flow simulation along the OIA/JKB2n/X45 radar transect (see red dotted line in Fig. 1 for location) (TOP) Various inferred parameters (plain lines) as well as their 15 and 85 percentiles (dashed lines). From top to bottom: average surface accumulation rate, geothermal heat flux, p+1 parameter of the velocity profile, average basal melting, bottom age 60 m above bedrock, height above bed of the 1.5 Myr isochrone and resolution of the 1.5 Myr isochrone. (BOTTOM) Modeled age (in colour scale, white is for ages older than 1.5 Myr), together with observed isochrones (in white) and bed (in thick black).** Note the two main Oldest Ice candidates at distance 25 km (NP) and at distance 75 km (LDCP).

Figure 4 right bottom: The authors argue that the temporal resolution of 1.5Ma ice is about 10 ka/m, which is hard to read from the figure. Change the color scale.

We changed the color scale of Figure 4 right bottom, so that a 10 kyr/m resolution corresponds to the light blue color:

[Figure]

Figure 5 right top: please revise the color scale. Now most values are shown with blue-ish color so the figure shows little information.

[Figure]

Figure 5 for geothermal flux: see my comment for line 236.

[revised manuscript text omitted]